# NET Formation in Systemic Lupus Erythematosus: Changes during the COVID-19 Pandemic

**DOI:** 10.3390/cells11172619

**Published:** 2022-08-23

**Authors:** Jasmin Knopf, Johanna Sjöwall, Martina Frodlund, Jorma Hinkula, Martin Herrmann, Christopher Sjöwall

**Affiliations:** 1Department of Internal Medicine 3—Rheumatology and Immunology, Friedrich-Alexander-Universität Erlangen-Nürnberg (FAU) and Universitätsklinikum Erlangen, 91054 Erlangen, Germany; 2Deutsches Zentrum für Immuntherapie (DZI), Friedrich-Alexander-Universität Erlangen-Nürnberg (FAU) and Universitätsklinikum Erlangen, 91054 Erlangen, Germany; 3Department of Biomedical and Clinical Sciences, Division of Inflammation and Infection/Infectious Diseases, Linköping University, SE-581 85 Linköping, Sweden; 4Department of Biomedical and Clinical Sciences, Division of Inflammation and Infection/Rheumatology, Linköping University, SE-581 85 Linköping, Sweden; 5Department of Biomedical and Clinical Sciences, Division of Molecular Medicine and Virology, Linköping University, SE-581 85 Linköping, Sweden

**Keywords:** COVID-19, SARS-CoV-2, systemic lupus erythematosus (SLE), neutrophils, neutrophil extracellular traps (NETs)

## Abstract

The severity of the coronavirus disease in 2019 (COVID-19) is strongly linked to a dysregulated immune response. This fuels the fear of severe disease in patients with autoimmune disorders continuously using immunosuppressive/immunomodulating medications. One complication of COVID-19 is thromboembolism caused by intravascular aggregates of neutrophil extracellular traps (NETs) occluding the affected vessels. Like COVID-19, systemic lupus erythematosus (SLE) is characterized by, amongst others, an increased risk of thromboembolism. An imbalance between NET formation and clearance is suggested to play a prominent role in exacerbating autoimmunity and disease severity. Serologic evidence of exposure to SARS-CoV-2 has a minor impact on the SLE course in a Swedish cohort reportedly. Herein, we assessed NET formation in patients from this cohort by neutrophil elastase (NE) activity and the presence of cell-free DNA, MPO-DNA, and NE-DNA complexes and correlated the findings to the clinical parameters. The presence of NE-DNA complexes and NE activity differed significantly in pre-pandemic versus pandemic serum samples. The latter correlated significantly with the hemoglobin concentration, blood cell counts, and complement protein 3 and 4 levels in the pre-pandemic but only with the leukocyte count and neutrophil levels in the pandemic serum samples. Taken together, our data suggest a change, especially in the NE activity independent of exposure to SARS-CoV-2.

## 1. Introduction

Infections caused by the severe acute respiratory syndrome coronavirus 2 (SARS-CoV-2) have caused a still ongoing worldwide pandemic, claiming the lives of more than 6.3 million people to date [1]. Advanced age and comorbidities have been identified as major risk factors for developing severe disease, suggesting that the weakened immune system in these at-risk groups is a major contributor [2]. Patients who are constantly taking immunosuppressive or immunomodulating medications for underlying autoimmune disorders are, therefore, increasingly afraid of developing the severe coronavirus disease from 2019 (COVID-19), especially since conflicting results have been published on the susceptibility to SARS-CoV-2 infection and the risk of severe disease in patients with autoimmune diseases [3]. Additionally, a subset of patients with moderate to severe COVID-19 developed clinical features similar to those seen in autoimmune rheumatic diseases (e.g., autoantibodies) [4]. Type I interferons (IFNs) are central to antiviral immunity, and a deficiency of type I IFNs in the blood of critically ill patients was recently associated with a persistent blood viral load and an exacerbated inflammatory response [5]. However, in conflicting reports, patients with severe COVID-19 also exhibited a robust and sustained type I IFN response that contributed to organ damage [6,7,8]. Like this subset of patients with the SARS-CoV-2 infection, patients with systemic lupus erythematosus (SLE) suffer from an autoimmune disease characterized, amongst others, by a dysregulated type I IFN response [9]. Inflammation in SLE is frequently marked by elevated type I IFN levels; however, whether these are potentially protective needs further evaluation. Despite a recent study of Swedish SLE patients with longitudinal follow-ups revealing only a minor impact of SARS-CoV-2 exposure on the course of SLE [10], studies on the impact of the SARS-CoV-2 infection in the context of SLE are still scarce.

A major clinical feature of severe COVID-19 is thromboembolism, which occurs due to vascular occlusion and disturbed microcirculation. We and others have recently shown that this is caused by the intravascular aggregation of neutrophil extracellular traps (NETs), leading to the rapid occlusion of the affected vessels, especially the capillaries [11,12,13,14]. NETs are composed of a chromatin scaffold equipped with various granular and cytoplasmic neutrophilic enzymes necessary for the killing of pathogens and the degradation of pro-inflammatory cytokines and chemokines. However, an unbalance in NET formation is also implicated in many rheumatic and autoimmune diseases, and NETs reportedly occlude vessels and ducts [15,16]. This disbalance between NET formation and degradation is also increasingly acknowledged to play a major part in the pathophysiology of coagulopathy, inflammation, immune-mediated thrombotic events, and organ damage in patients with severe COVID-19 [17]. Like the SARS-CoV-2 infection, SLE is not only characterized by an imbalanced type I IFN response but also by an increased risk of thromboembolism and robust activation of the classical complement pathway. Furthermore, an imbalance between NET formation and clearance in patients with SLE has been suggested to play a prominent role in exacerbating autoimmunity and disease severity [18]. A failed degradation of NETs in patients with SLE correlates with more active disease and lower levels of complement proteins C3 and C4 since NETs may activate complement in vitro [19]. In addition, genetic alterations have been shown to affect NET formation in patients with SLE [20]. Finally, NETs purified from SLE patients were able to stimulate an ex vivo IgG2 isotype class switch, concurring with the renal lesions characteristic of lupus nephritis and the release of soluble IgG2 from SLE patients’ B cells [21]. Neutrophils from subjects with SLE are reportedly more prone to release NETs, and these NETs can potentially activate plasmacytoid dendritic cells to produce type I IFNs [22,23].

Since there is, to our knowledge, no data available on the impact of NET formation in SLE patients with serological signs of exposure to the SARS-CoV-2 infection and on changes in NET formation of the same patient over time, we herein examined four NET parameters: (1) the presence of cell-free DNA, (2) NE-DNA complexes, (3) MPO-DNA complexes, and (4) the activity of NE in the sera of the aforementioned Swedish SLE cohort [10].

## 2. Materials and Methods

### 2.1. Subjects

The study population consisted of 100 patients (83 women, 17 males) with established SLE with regular physical visits from March 2020 to January 2021 to the rheumatology unit at the Linköping University Hospital, Sweden, as described in detail in Sjöwall et al. [10]. All patients fulfilled the 1982 American College of Rheumatology (ACR) and/or the 2012 Systemic Lupus International Collaborating Clinics (SLICC) classification criteria for SLE. The patients had previously been included in the prospective follow-up program, KLURING (a Swedish acronym for *Clinical LUpus Register in North-Eastern Gothia*), at the Department of Rheumatology, Linköping University Hospital [24]. The clinical disease activity of these patients was assessed using an SLE disease activity index-2000 (SLEDAI-2K) with the exclusion of the laboratory items for complement consumption and anti-double-stranded DNA binding (mSLEDAI) [25].

As described in Sjöwall et al. [10], the participating subjects donated blood consecutively during their regular visit during the pandemic prior to vaccination. They also had done likewise at another visit to the rheumatology unit prior to the pandemic. Thus, a corresponding pre-pandemic serum sample (from August 2015 to November 2019) was available from each participating subject. All serum samples were stored at −70 °C until analysis. Detailed characteristics of the study population and their pharmacotherapy are shown in Table 1.

### 2.2. Routine Laboratory Measurements and Autoantibody Analyses

The collection of data from routine laboratory measurements and autoantibody analyses has been described elsewhere [10].

### 2.3. SARS-CoV-2 PCR Assay and SARS-CoV-2 Antibody Assays

Detection of SARS-CoV-2 by PCR and SARS-CoV-2 antibodies (IgG, IgA, and IgM) have been described elsewhere [10].

### 2.4. Assessment of NET Markers

To assess the activity of neutrophil elastase (NE) in the serum of patients with SLE, 10 µL of serum and a 100 µM fluorogenic NE substrate (MeOSuc-AAPV-AMC, sc-201163, Santa Cruz Biotechnology Inc., Dallas, TX, USA) per well was added in a PBS in a 384-well plate (#3680, Corning Inc., Corning, NY, USA). The conversion of the fluorogenic substrate (Ex.: 360 nm, Em.: 465 nm) by NE in the serum was assessed by an increase in the mean fluorescence intensity (MFI) using a Tecan Infinite F200Pro (Tecan Group Ltd., Männedorf, Switzerland) for 20 h at 37 °C. The first value was subtracted as the baseline from the endpoint measurement taken after 20 h. The amount of cell-free DNA in the serum of patients was determined using the Quant-iT™ PicoGreen™ dsDNA Assay-Kit (P11496, Thermo Fisher Scientific Inc., Waltham, MA, USA) according to the manufacturer’s instructions. Briefly, 5 µL of serum was diluted to 100 µL with the TE buffer supplied before 100 µL of the diluted PicoGreen™ reagent was added. After incubation for 5 min at RT on a shaker, the fluorescence (Ex.: 485 nm, Em.: 535 nm) was measured on a Tecan Infinite F200Pro. Concentrations were calculated using the supplied DNA standard. The presence of myeloperoxidase (MPO)-DNA and NE-DNA complexes in the serum was measured using ELISA modified from the previously reported [26,27]. Briefly, Nunc MaxiSorp™ 96-well plates (#442404, Thermo Fisher Scientific Inc.) were coated with 100 µL/well of an anti-MPO antibody (ab68672, RRID: AB_1658868, Abcam, Cambridge, UK) or anti-NE antibody (MAB91673, RRID: AB_ AB_2920862, R&D Systems, Minneapolis, MN, USA) overnight at +4 °C. The next day, the plates were washed thrice with a wash buffer (0.05% Tween-20 in PBS) and blocked with 200 µL/well of 3% BSA in PBS for 2 h at RT, shaking. After another washing step, 40 µL of the serum was added to each well and incubated for 2 h at RT, shaking. Then, 100 µL/well of an anti-DNA POD, diluted at 1:40 in an incubation buffer, both from a Cell Death Detection Kit (11544675001, Merck KGaA, Darmstadt, Germany), was added after another washing step for 90 min at RT in the dark and with shaking. After one last washing step, the reaction was developed by incubating 100 µL/well of a TMB Substrate (#421101, BioLegend, San Diego, CA, USA) for 30 min at RT. The reaction was stopped by the addition of 50 µL/well of 25% of H_2_SO_4_ (#122448.1211, AppliChem GmbH, Darmstadt, Germany), and the absorbance was read at 450 nm (with a reference at 620 nm) using a Sunrise microplate reader (Tecan Group Ltd.). To compare the results from the different plates, a standard serum was used on each plate to calculate a plate variation factor.

### 2.5. Statistics

Statistical correlations were calculated using IBM SPSS Statistics Version 28.0.0.0 (190) (IBM, Armonk, NY, USA). Graphs and figures were created using GraphPad Prism Version 9.0.2 (161) (GraphPad Software, LLC, San Diego, CA, USA) and Microsoft Office Professional Plus 2019 (Microsoft, Redmond, WA, USA).

### 2.6. Ethical Considerations

This study was carried out in accordance with the Declaration of Helsinki. Written informed consent was obtained from all participants. The study protocol was approved by the Regional Ethics Review Board in Linköping (Decision number M75-08/2008).

## 3. Results

To assess changes in the formation of NETs between the serum samples taken pre-pandemic and during the pandemic, we analyzed the activity of the NE, the amount of cell-free DNA, and the presence of NE-DNA and MPO-DNA complexes. Compared to the pre-pandemic serum samples, only the activity of NE and the presence of NE-DNA complexes were significantly different in the pandemic samples (Figure 1a,b,e). No statistical difference was observed in the amount of cell-free DNA (Figure 1c) and the presence of MPO-DNA complexes (Figure 1d), even though there was a trend towards higher amounts of MPO-DNA complexes in the pandemic samples (Figure 1e).

Next, we examined if these changes in the NE activity also correlated with other clinical and laboratory variables. All parameters used for comparisons are summarized in Table 1. Out of these, the levels of hemoglobin (Figure 2a), white blood cell count (WBC, Figure 2b), neutrophils (Figure 2c), lymphocytes (Figure 2d), complement protein 3 (C3, Figure 2e), and complement protein 4 (C4, Figure 2f) correlated significantly with the activity of NE in the serum of the pre-pandemic cohort. Interestingly, despite higher NE activity in the pandemic samples (Figure 1a,e), the activity only correlated with the WBC (Figure 2g) and neutrophils (Figure 2h) in these samples.

To further assess the impact of the exposure to SARS-CoV-2, we divided the samples further into antibody and/or PCR (=SARS-CoV-2-) positive and SARS-CoV-2-negative samples and performed correlation analyses with the same parameters (Table 1) again. Since the PCR testing of the individuals with symptoms of infection was not introduced in Sweden in the routine for the general population until June 2020 [28], only four of the patients in the pandemic cohort had a PCR-confirmed infection. However, 36 patients in total (34 subjects without PCR confirmation and two of the four PCR-confirmed patients) had antibodies of one or more isotypes against the SARS-CoV-2 spike protein in their sera. Therefore, we considered a total of 38 patients from the pandemic cohort as SARS-CoV-2-positive, as indicated in Table 2. As described in Table 2, we detected antibodies of different isotypes against SARS-CoV-2 in the sera of 41 of the study subjects already before the onset of the pandemic. Changes in the distribution and levels of the antibody isotypes between the pre-pandemic cohort and the pandemic cohort are also depicted in Appendix A. Of note, of the 41 individuals already seropositive before the onset of the pandemic, 28 were still positive in the pandemic cohort but with changes in the levels and distribution of the antibody isotypes, as indicated in Appendix A.

The NE activity of the SARS-CoV-2-negative pre-pandemic samples correlated only with the levels of hemoglobin (Figure 3a) and plasma creatinine (Figure 3b), whereas in the SARS-CoV-2-positive samples of this cohort, the NE activity correlated with the WBC (Figure 3c) and neutrophils (Figure 3d), similar to the samples from the pandemic cohort (Figure 2g,f). Of note, the MPO-DNA complexes in the SARS-CoV-2-positive serum samples from the pre-pandemic period were also associated with positivity for IgA antibodies against SARS-CoV-2 (data not shown).

Interestingly, in the SARS-CoV-2-negative samples, the NET markers did not correlate significantly with any of the other clinical parameters, whereas in the SARS-CoV-2-positive samples, the same significant positive correlation between the NE activity and WBC (Figure 3e) and neutrophils (Figure 3f) was observed. Additionally, the amount of cell-free DNA detected in these serum samples was associated inversely with malar rash (ACR1, data not shown). Of note, the NE activity, but no other NET parameter, is significantly higher in the pandemic compared to the pre-pandemic samples independent of serological signs of exposure to SARS-CoV-2, as shown in Appendix A.

## 4. Discussion

In the present study, we aimed to assess changes in active NET formation in patients with SLE who showed serological signs of exposure to SARS-CoV-2. Both diseases share many common features, such as an imbalanced type I IFN response and robust activation of complement. It is also becoming increasingly evident that neutrophils, and especially the disbalance between NET formation and degradation, play a major role in the pathologies of these diseases [17,29]. To our knowledge, we are the first to report a change in the activity of the NET-associated enzyme NE and the presence of NE-DNA complexes in the sera of patients with SLE taken during the pandemic and compared to pre-pandemic samples of the same patients. It would be, of course, interesting to see if these changes of NET parameters in the serum between two time points can only be found in patients with SLE or also in patients with other (autoimmune) diseases or healthy individuals. Unfortunately, since it is very difficult to get hold of such samples from healthy controls, this is a limitation of our study. Since both diseases, SLE and COVID-19, are associated with an imbalance in NET formation and degradation, our results raise the question of if increased NET formation is a result of the COVID-19 infection or an increased disease activity of SLE/increased autoimmunity due to the infection with SARS-CoV-2. It is conceivable that both possibilities contribute to the increased NET formation and that it is rather an individual reaction towards the infection with SARS-CoV-2. This is indicated by the quite heterogenous changes in the disease activity (mSLEDAI) of the patients with serological signs of exposure, pre-pandemic vs. pandemic. Of these patients, 25 had the same disease activity index as before the pandemic; four had a higher index, and nine had a lower index. However, this also goes in line with the changes in the dose of DMARDs/prednisolone or the type of DMARDs, as indicated in Appendix A.

Intriguingly, the other NET-markers examined, cell-free DNA and MPO-DNA complexes, did not show a significant difference in these sera. Cell-free DNA in healthy individuals is usually of a hematopoietic origin [30]. Since the description of NETs, these are considered the prime origin of cell-free DNA and a valuable diagnostic marker [31]. Nevertheless, cell-free DNA is also released from dying cells/affected tissue in pathological conditions, e.g., it is proposed as a biomarker for cancer [32,33]. In the context of SLE, high levels of circulating DNA are considered a biomarker for active disease [34], especially in association with renal involvement [35]. Since most patients in the cohort included in this study were in remission or had low clinical disease activity (the mean mSLEDAI scores were 1.34 and 0.82, respectively, [see Table 1] and see [10] for more disease activity variables) that did not change significantly during the two sampling time points, it is conceivable that the levels of cell-free DNA indicative of active disease are similar between the pre-pandemic and pandemic samples.

Interestingly, a total of 41 samples obtained during the pre-pandemic period showed seropositivity for SARS-CoV-2. It is now well accepted that antibodies against human seasonal coronaviruses show a cross-reactivity against SARS-CoV-2 antigens and that these show a certain amount of neutralizing activity [36,37]. By using a flow-cytometry-based method, Ng et al. demonstrated that SARS-CoV-2 uninfected individuals had cross-reactive antibodies against the S1 subunit of the spike protein, whereas individuals infected with SARS-CoV-2 developed antibodies against both subunits [38]. The antibodies of a small proportion of uninfected individuals with antibodies against the S2 subunit were of the IgG serotype but not IgA or IgM. However, cross-reactive IgA against the S1 subunit was also detected in the saliva of uninfected individuals and correlated negatively with age [39], possibly because children and adolescents have generally higher infection rates with seasonal coronaviruses than adults [38]. Since the mean age of our cohort pre-pandemic and pandemic was 48.7 and 51.3 years, respectively [10], this explanation seems unlikely in our case. Further possible explanations are the interference with autoantibodies in the immunoassays or unspecific immunoglobulin responses. It was already shown that background diseases, such as malaria, dengue fever, or schistosomiasis, show a high cross-reactivity with the COVID-19 antibody detecting test [40]. The first indications of cross-reactions of SARS-CoV antigens with autoantibodies in autoimmune diseases, such as SLE, were already observed following the SARS pandemic in 2002/2003 [41]. Additionally, it is also conceivable that patients with SLE, a disease characterized by the broad production of autoantibodies, in general, might be more prone to develop antibodies against various antigens.

MPO is a granular neutrophilic enzyme regulating the formation of NETs by driving chromatin decondensation [42]. The ELISA for the detection of MPO-DNA complexes was developed by Kano et al. based on an earlier description of Kessenbrock et al. as a reliable assay for the detection of NETs in clinical settings [26,27]. Blood MPO-DNA levels are discussed as a biomarker of the early response against the SARS-CoV-2 infection [43], and elevated levels have also been detected in the sera of patients with IgA vasculitis [44]. In general, the increased presence of circulating NETs (remnants) detected by ELISA was recently described for many autoimmune conditions and infections [45,46]. Lood et al. and Bruschi et al. have found increased MPO-DNA plasma/serum levels in patients with SLE and associated these with higher disease activity/lupus nephritis [47,48]. Since disease activity in this patient cohort was well controlled, it is not surprising that this NET parameter showed no significant difference between the two sampling time points. However, there was a trend towards higher levels of MPO-DNA complexes in the pandemic serum samples. Additionally, a recent paper published by Hayden et al. suggests that the ELISA for the detection of MPO-DNA complexes in human plasma is error-prone and highly questionable regarding the specificity of NET detection in vivo [49].

Taken together, the activity of NE and the presence of NE-DNA complexes were significantly higher in serum samples taken during the pandemic (prior to vaccination) compared to pre-pandemic samples. Significant associations with clinical parameters were limited, but correlations of NE activity with the WBC and neutrophils were observed, especially in the pandemic samples that had signs of exposure to SARS-CoV-2 (the presence of antibodies and/or PCR-confirmed infection). This study is also, to our knowledge, the first time that changes in the NET parameters NE- and MPO-DNA complexes, cell-free DNA, and NE activity in the serum were assessed within the same patient over time. Since the exposure to SARS-CoV-2 did not seem to have a major impact on the disease severity in this cohort of SLE patients, further research is needed to elucidate the reason for the significantly higher NE activity and levels of NE-DNA complexes in the pandemic serum samples.

## Figures and Tables

**Figure 1 cells-11-02619-f001:**
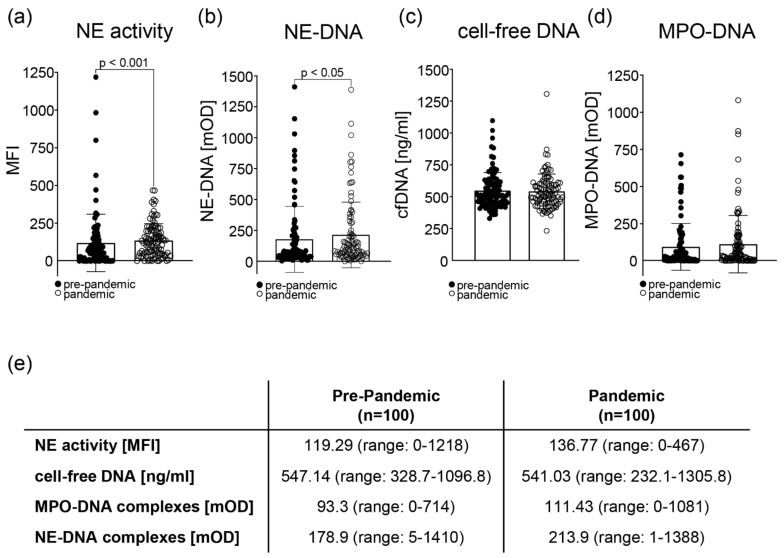
Markers for neutrophil extracellular traps (NETs) in the sera of patients with SLE pre-pandemic compared to the pandemic. (**a**) The activity of neutrophil elastase (NE) was assessed by the increase in the mean fluorescence intensity (MFI) after the conversion of a specific fluorescent substrate. The activity of the NE was significantly higher (*p* < 0.001) in the pandemic sera. (**b**) The presence of NE-DNA complexes was significantly different (*p* < 0.05) between the pre-pandemic and pandemic samples, as analyzed by ELISA. (**c**) The presence of cell-free DNA in the serum of patients was measured by a Quant-iT PicoGreen dsDNA Assay Kit, and no significant differences were observed between the different time points. (**d**) The presence of MPO-DNA complexes was analyzed by ELISA, showing no significant differences between the pre-pandemic and the pandemic cohorts. (**e**) Mean values and ranges for all three NET parameters were analyzed (NE activity, cell-free DNA, and MPO-DNA complexes) in the sera of SLE patients: pre-pandemic vs. pandemic. Statistical significance was calculated using the Wilcoxon matched-pairs signed rank test.

**Figure 2 cells-11-02619-f002:**
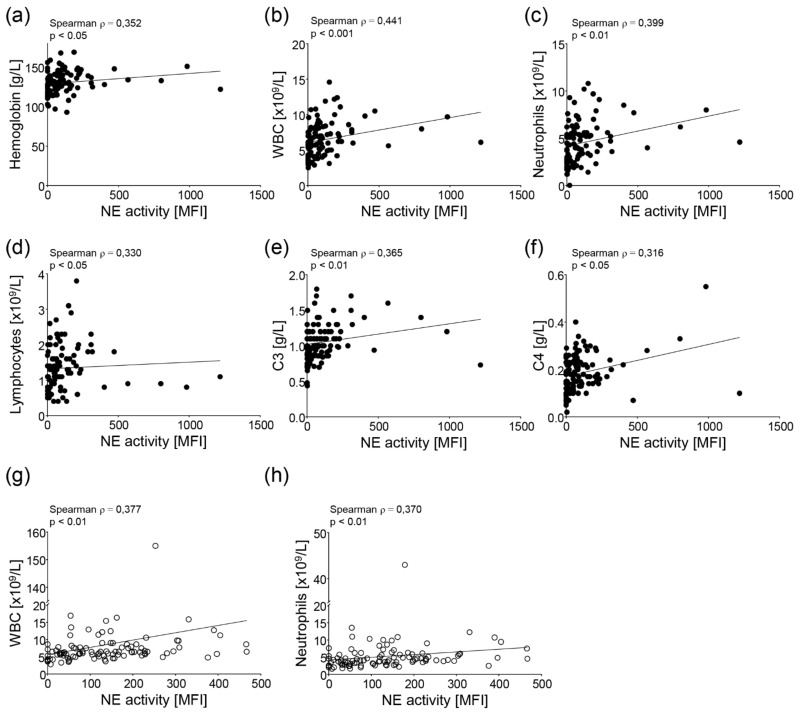
Correlation of NE activity with other laboratory variables associated with the disease activity of the SLE patients. The activity of neutrophil elastase (NE) correlated significantly with the (**a**) hemoglobin concentration, (**b**) white blood cell count (WBC), (**c**) neutrophil count, (**d**) lymphocyte count, (**e**) levels of complement protein 3 (C3), and (**f**) levels of complement protein 4 (C4) in the pre-pandemic samples. In the pandemic samples, the activity of neutrophil elastase (NE) correlated significantly with the (**g**) WBC and (**h**) neutrophil count. A Spearman rank correlation with a Bonferroni correction for multiple comparisons was used for statistical testing.

**Figure 3 cells-11-02619-f003:**
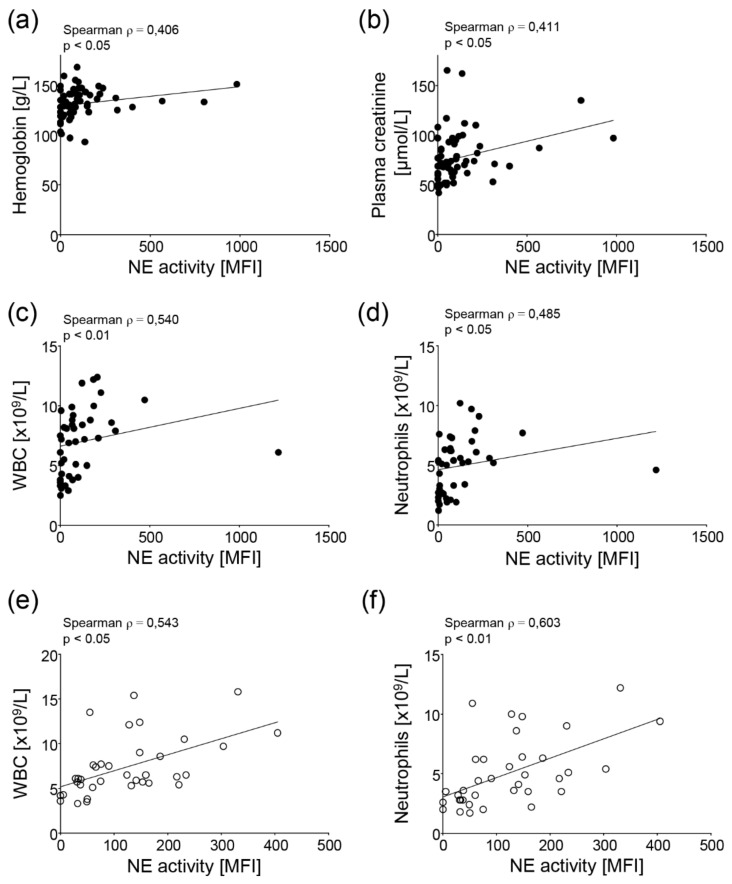
Correlation of the NE activity with other laboratory variables associated with the disease activity of the SARS-CoV-2-negative and -positive SLE patients. In the pre-pandemic SARS-CoV-2--negative (no detectable antibodies against severe acute respiratory syndrome coronavirus type 2 (SARS-CoV-2) SLE patients, the NE activity correlated with (**a**) hemoglobin and (**b**) plasma creatinine levels. In the sera of patients with detectable antibodies against SARS-CoV-2 and/or PCR-confirmed infection (SARS-CoV-2-positive) from the pre-pandemic period, the NE activity correlated with the (**c**) white blood cell count (WBC) and (**d**) neutrophil counts. The same positive correlations of NE activity with the (**e**) WBC and (**f**) neutrophil counts were observed in the sera of the SARS-CoV-2-positive SLE patients obtained during the pandemic. A Spearman rank correlation with a Bonferroni correction for multiple comparisons was used for statistical testing.

**Table 1 cells-11-02619-t001:** Summary of the clinical parameters and pharmacotherapies used for correlation analyses. Data are given as mean values with range unless stated otherwise.

	Pre-Pandemic (*n* = 100)	Pandemic (*n* = 100)
Clinical Parameters		
Smoking former/ongoing (*n*)	31/11	44/5
BMI	26.75 (range: 17.04–42.67)	27.06 (range: 17.07–45.73)
Blood group 0 (*n*)	38	38
Blood group A (*n*)	37	37
Blood group B (*n*)	11	11
Blood group AB	4	4
Rh+ (*n*)	78	78
Rh− (*n*)	12	12
Hemoglobin	130.7 (range: 93–169)	130.2 (range 102–169)
Erythrocyte sedimentation rate (ESR) [mm/h]	20 (range: 2–108)	18.3 (range: 1–106)
WBC	6.5 (range: 2.5–14.6)	8.5 (range: 2.8–155)
Platelet count	253.3 (range: 1.9–500)	250.0 (range: 102–508)
Neutrophil count	4.6 (range: 0.02–10.8)	5.3 (range: 1.6–43)
Lymphocyte count	1.3 (range: 0.4–3.8)	1.5 (range: 0.3–4.5)
Plasma creatinine	73.9 (range: 31–165)	77.4 (range: 40–527)
Plasma albumin	-	39.1 (range: 28–50)
C-reactive protein (CRP) [mg/L]	6.8 (range: 2.5–172)	6.1 (range: 2.5–166)
Complement protein 3 (C3) [g/L]	1.06 (range: 0.4–1.8)	0.98 (range: 0.5–2)
Complement protein 4 (C4) [g/L]	0.19 (range: 0.02–0.55)	0.18 (range: 0.02–0.53)
Malar rash (ACR1) (*n*)	33	35
Discoid rash (ACR2) (*n*)	11	12
Photosensitivity (ACR3) (*n*)	48	48
Oral ulcers (ACR4) (*n*)	14	14
Arthritis (ACR5) (*n*)	78	80
Serositis (ACR6) (*n*)	29	29
Renal disorder (ACR7) (*n*)	34	34
Neurologic disorder (ACR8) (*n*)	10	10
Hematologic disorder (ACR9) (*n*)	58	62
Immunological disorder (ACR10) (*n*)	57	61
Antinuclear antibody (ACR11) (*n*)	99	99
mSLEDAI (score)	1.34 (range: 0–13)	0.82 (range: 0–22)
Anti-SSA/Ro52 Antibody levels [U/mL]	50.6 (range: 2–246)	47.2 (range: 0–254)
Anti-SSA/Ro60 Antibody levels [U/mL]	35.6 (range: 0–149)	34.2 (range: 0–178)
Anti-SSB/La Antibody levels [U/mL]	21.8 (range: 0–151)	19.5 (range: 0–150)
Anti-Sm/RNP Antibody levels [U/mL]	6.5 (range: 0–218)	2.8 (range: 0–75)
Anti-U1RNP Antibody levels [U/mL]	34.5 (range: 0–300)	27.6 (range: 0–327)
Anti-dsDNA Antibody levels [U/mL]	95.95 (range: 2–900)	101.28 (range: 0–1081)
**Pharmacotherapy**		
Immunosuppressives		
Prednisolone dose [mg/day]	4.5 (range:0–30)	3.5 (range: 0–15)
Azathioprine (*n*)	8	8
Methotrexate (*n*)	8	9
Mycophenolate mofetil (*n*)	21	23
Other		
Hydroxychloroquine (*n*)	77	73

**Table 2 cells-11-02619-t002:** Distribution of the COVID IgG antibody isotypes among the two cohorts.

	Pre-Pandemic (*n* = 41)	Pandemic (*n* = 38)
COVID Parameters		
COVID IgG Antibody positive (*n*)	4	8
COVID IgA Antibody positive (*n*)	31	30
COVID IgM Antibody positive (*n*)	13	9
COVID IgG + IgA Antibody positive (*n*)	0	5
COVID IgG + IgM Antibody positive (*n*)	0	1
COVID IgA + IgM Antibody positive (*n*)	7	5
**COVID PCR confirmed**	-	4
COVID IgG Antibody positive (*n*)	-	2
COVID IgA Antibody positive (*n*)	-	1
COVID IgM Antibody positive (*n*)	-	0
COVID IgG + IgA Antibody positive (*n*)	-	1
COVID IgG + IgM Antibody positive (*n*)	-	0
COVID IgA + IgM Antibody positive (*n*)	-	0
Duration of COVID symptoms (PCR-confirmed cases; days)	-	36.5 (range: 6–96)

## Data Availability

The raw data supporting the conclusions of this article will be made available by the authors without undue reservation.

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
