# Peer review of "NET Formation in Systemic Lupus Erythematosus: Changes during the COVID-19 Pandemic"

_cells, 2022, doi:10.3390/cells11172619_

Round 1
Reviewer 1 Report
I have read the extensively revised manuscript and found it to be greatly improved and for the most part addressed my previous concerns. I have no further concerns other than that there are sentences with a word missing or the wording is awkward due to the choice of words.
Author Response
We have revised the manuscript, included missing words and rephrased some sentences.
Reviewer 2 Report
The authors describe the NET formation in SLE patients during COVID19-pandemic compared to pre-pandemic in Swedish SLE patients. The results can be interesting to readers, however, several points should be more clearly discussed. Specific comments are listed below:
1. I wonder whether therapeutic agents influenced in the development of NETosis in SLE patients. The SLE patients included received unchanged agents during observation (PSL doses were found in the Table)? The treatment in SLE patients (immunosuppressants and/or HCQ) should be described in the Table.
2. Table 1, why the SLE symptoms are vacant in pre-pandemic, regardless of mean SLEDAI scores were given?
3. It is difficult to assess the real influence of COVID19 in NETosis because pre-pandemic SLE patients also had COVID immunoglobulins (Igs) and the number of patients with COVID Igs did not significantly increased during pandemic. I wonder the increase of NETosis really the result of COVID19 infection itself, or increased activity of SLE (increased autoimmunity) due to COVID19. Which is more likely to explain the results in this study?
Author Response
The authors describe the NET formation in SLE patients during COVID19-pandemic compared to pre-pandemic in Swedish SLE patients. The results can be interesting to readers, however, several points should be more clearly discussed. Specific comments are listed below:
- I wonder whether therapeutic agents influenced in the development of NETosis in SLE patients. The SLE patients included received unchanged agents during observation (PSL doses were found in the Table)? The treatment in SLE patients (immunosuppressants and/or HCQ) should be described in the Table.
We included now additional information on pharmacotherapy of the patients in Table 1. There were no significant correlations between treatment and any of the NET parameters assessed.
- Table 1, why the SLE symptoms are vacant in pre-pandemic, regardless of mean SLEDAI scores were given?
We apologize for the misunderstanding. The ACR criteria are not SLE symptoms at a given timepoint – they are also cumulative. However, in Table 1, we have added the requested information relevant for the time point of sampling of the pre-pandemic samples.
- It is difficult to assess the real influence of COVID19 in NETosis because pre-pandemic SLE patients also had COVID immunoglobulins (Igs) and the number of patients with COVID Igs did not significantly increased during pandemic. I wonder the increase of NETosis really the result of COVID19 infection itself, or increased activity of SLE (increased autoimmunity) due to COVID19. Which is more likely to explain the results in this study?
It is true that number of patients with COVID Igs did not significantly increase during the pandemic. This is probably because these patients were extra cautious in their daily life as at-risk person to avoid any infections (personal observations from C. Sjöwall). It is really difficult to say if the increase in NET formation is the result of the infection itself or due to increased autoimmunity due to the infection. We would speculate that both possibilities contribute to the increased NET formation and that is rather an individual reaction towards the infection. Looking at the individual changes of disease activity (mSLEDAI) of the patients with serological signs of COVID-19 exposure, the cohort is quite heterogenous. Of the SARS-CoV-2 positive patients from the pandemic, 25 had the same disease activity index as before the pandemic, 4 had a higher index and 9 a lower index. However, this goes, in some cases, in line also with changes in the dose of immunosuppressants/HCQ and type of DMARD as shown in the new Supplementary Table 2.
Round 2
Reviewer 2 Report
The revised manuscript shows substantial improvement. I have no further comments.
This manuscript is a resubmission of an earlier submission. The following is a list of the peer review reports and author responses from that submission.
Round 1
Reviewer 1 Report
An manuscript entitled Net formation in systemic lupus erythematosus: Changes during the COVID-19 pandemic has been submitted for publication in the journal Cells by Jasmin Knopf et al. This study is focused on assessing Net formation by neutrophils in SLE patients prior to and during the COVID-19 pandemic time frame of March 2020 to January 2021. The authors asked the question as to whether exposure to SARS-CoV-2 affected NET formation in their SLE cohort based on serological exposure. The authors looked a for different parameters of NET formation and correlated their assay findings with the corresponding clinical findings. The authors conclude from their study that there is a significant difference be pre-pandemic and pandemic samples for two parameters: NE-DNA complexes and NE activity. However, when all data is taken into account the data suggest there is a change in NE activity that is independent of SARS-CoV-2 exposure.
The study is straight forward and its strength is that the same 100 SLE study subjects were able to be studied both pre-pandemic and as COVID-19 pandemic exposed. An additional strength is that 4 validated assays measuring different parameters of NET formation were assessed in conjunction with routinely collected clinical assessment data.
Concerns:
1. The study was based solely on the 100 SLE subjects and did not make a comparison to healthy control subjects over the same period of time the study was conducted. Would one have seen similar changes in a healthy, non-SLE subjects.
2. A significant amount of the introduction is devoted to SLE and SARS-CoV-2 infection, yet the data analysis is presented only from the context of the 100 subjects in the study regardless of their serological status. The authors indicate in the introduction that the assessment is made on the basis of “serological signs of exposure to SARS-CoV-2”. Yet in table 1, the authors do not indicate how many of the pandemic samples had serological evidence of viral infection as it is not clear if, for example, those that are IgA positive are also IgG, IgM and/or PCR positive. The number of pandemic samples with evidence of infection should be clearly stated and whether subjects were positive for one or more markers of infection. In essence only two cohort subjects in the 100 were confirmed to be SARS-CoV-2 positive. This low confirmed SARS-CoV-2 number in light of the pre-pandemic sample sero-positivity data weakens the study and confounds the interpretation of the data.
3. In the beginning of the discussion, it was stated that 59 of the 100 pre-pandemic subject samples had evidence of coronavirus infection based on the assays used. This information does not match the numbers in Table 1 and is further confusing because it is not clear (again) which if any of the samples are positive for more than one serological marker or not. Are the same individuals positive pre-pandemic also positive post-pandemic or did status change? Did antibody titers go up in the pandemic samples if they were positive in the pre-pandemic samples that might indicate SARS-CoV-2 infection? These issues need clarification in order to properly interpret the findings presented in Figure 1 and the subsequent Figures 2 and 3.
4. In Figure 1 if one separates out the coronavirus positive samples pre-pandemic and pandemic from the negative samples are there significant differences?
5. In many ways this is a study of whether overall coronavirus infection has an impact on SLE clinical parameters over time as opposed to a SARS-CoV-2 specific response. In the end the authors conclude there is no association between NETs and SARS-CoV-2 infection. So not sure what this paper really contributes to the field or to the understanding of NET formation, SLE and SARS-CoV-2 infection.
Reviewer 2 Report
Knopf et al investigated NET formation in serum samples of patients with SLE pre-pandemic versus CVID-19 pandemic . They stated that NET formation will be correlated with clinical parameters.
Major issues:
Unfortunately, only 4 out of 100 patients had positive PCR for COVID 19. No correlation was done with clinical parameters.
The only correlation performed was with laboratory parameters such as hemoglobin, and leukocyte counts .
Also, the serology was positive even before the COVID 19 pandemic , making it hard to compare pre and pandemic time.
In the discussion- line 285- " higher levels of NE activity had no impact on disease severity - yet no description of disease severity is documented.